# Relationship between Lifestyle Determinants and Perceived Mental and Physical Health in Italian Nursery and Primary School Teachers after the COVID-19 Lockdown

**DOI:** 10.3390/jfmk9010033

**Published:** 2024-02-17

**Authors:** Antonino Scardina, Garden Tabacchi, Ewan Thomas, Giovanni Angelo Navarra, Luca Petrigna, Giovanni Caramazza, Antonio Palma, Marianna Bellafiore

**Affiliations:** 1Sport and Exercise Sciences Research Unit, Department of Psychology, Educational Science and Human Movement, University of Palermo, 90144 Palermo, Italy; antonino.scardina01@community.unipa.it (A.S.); garden.tabacchi@unipa.it (G.T.); ewan.thomas@unipa.it (E.T.); giovanniangelo.navarra@community.unipa.it (G.A.N.); giovanni.caramazza1@istruzione.it (G.C.); antonio.palma@unipa.it (A.P.); 2Department of Biomedical and Biotechnological Sciences, Section of Anatomy, Histology and Movement Science, School of Medicine, University of Catania, 95123 Catania, Italy; luca.petrigna@unict.it; 3Regional School Office of Sicily (USR Sicilia), 90144 Palermo, Italy

**Keywords:** Mediterranean diet, physical activity, body mass index, sedentary behavior, workplace, health, kindergarten

## Abstract

The COVID-19 lockdown and the consequent distance school learning made epochal changes in children’s lifestyles; however, little is known about the lockdown effects on school teacher habits. The aim of this observational study is to examine differences in the lifestyle of nursery (NS) and primary (PS) school teachers after one of the COVID-19 lockdowns and investigate the relationship between perceived physical/mental health and demographics, weight status and lifestyle determinants, such as adherence to the Mediterranean Diet (MD) and physical activity level (PAL). A total sample of 265 participants (49.22 ± 6.95 years) filled out an online information questionnaire and standardized questionnaires to collect data on the Physical Component Summary (PCS), Mental Component Summary (MCS), PAL and MD-adherence. A *t*-test or ANOVA analysis was used to assess differences between quantitative variables: Mann–Whitney U or Kruskal–Wallis tests for qualitative variables. Spearman’s correlations and multinomial logistic regression analyses were performed to identify categorical factors associated with classes of PCS and MCS. Teachers showed sufficient/high PAL, with a significantly higher level in the PS group (*p* < 0.05). This last showed a higher PCS score (*p* < 0.05). No differences between groups were found for MD-adherence, which was moderate/fair in both groups, and MCS, which was sufficient/good. Logistic regression showed that the only positive predictor of a high PCS was being physically active (OR 2.10, 95%CI 1.05–4.2, *p* < 0.05), while MCS was positively associated with normal weight status (OR 0.48, 95%CI 0.33–0.78, *p* = 0.020). This study highlights that PS teachers are more active than the NS group and perceive a higher physical health level. Mere predictors of physical and mental well-being are PA practice and being normal weight, respectively. This suggests that interventions to improve perceived health in this work category should be focused on the promotion of physical activity practice and on the maintenance of an optimal weight status.

## 1. Introduction

The COVID-19 pandemic and the consequent closing of schools with the inclusion of distance learning have made epochal changes in the lifestyles of children, affecting their physical and mental health. It has been demonstrated that in children aged 1 to 10 years, an increased use of digital technologies linked to distance learning has led to negative changes in sleep, emotions and behavior [1]. Such changes were also recorded among the teachers’ population, adversely affecting their lifestyle and physical and mental health [2,3,4,5,6]. Before the lockdown, school teachers already represented a category of workers characterized by chronic fatigue, physical inactivity, the presence of musculoskeletal disorders, voice disorders and common mental disorders [7,8]. The lockdown has affected, from a mental point of view, the perception of well-being concerning professional practice, suggesting the need to develop interventions to improve professional well-being [3,4], while from a physical point of view, the teacher population showed sedentary behaviors and high levels of overweight and obesity [5,9,10]. Eating habits have also undergone changes, with an increase in the consumption of unhealthy foods (sweets, soft drinks, sausages, frozen foods and salty foods) [10,11]. Physical exercise performed at home during the lockdown appears to have had a positive effect on physical well-being, but there are still conflicting opinions on the effects on mental well-being [9,12]. Given that a worker’s productivity is associated with their own level of physical and mental health, which, in turn, depends on multiple lifestyle determinants, including demographic and socioeconomic factors, education level, weight status, physical activity and workload [13,14], the health of teachers may affect the quality of schoolchildren’s education. An unhealthy lifestyle characterized by physical inactivity and incorrect eating habits is among the main risk factors for the development of chronic noncommunicable diseases (such as cardiovascular disease, cancer, chronic respiratory disease and diabetes) [15]. Specifically, sedentary behaviors (SBs) can show up at different times of daily life, during free time or at work [16]. SBs are defined as the time spent in sitting and/or reclining position during waking hours, with energy expenditure less than 1.5 metabolic equivalents (METs) [17]. Absenteeism (time away from work due to illness or disability) and presenteeism (reduced productivity at work) can result from consequences associated with SBs, impacting medical expenses and work productivity as well as the health of the individual [18,19]. The majority of daily sedentary time is spent in the work setting, where working adults spend almost half of their waking time (between 8 and 11 h per day) [20]. For example, office workers spend at least two-thirds of their working hours sitting [21], while workers such as teachers are considered “non-sedentary” during working hours because it has been estimated that they remain in an orthostatic position for 95 percent of working time [22,23]. However, this does not preclude this category of workers from performing sedentary activities during their free time, such as watching television and using computers or cell phones. In fact, some scientific evidence suggests low levels of physical activity (PA) and high levels of obesity in this category of workers [24,25]. Apparently, age has an influence; indeed, teachers with high sedentary behavior exhibit significantly lower age than others, and this could be explained by greater time spent in front of digital media, which are known as the set of means of communication that focus on digitized technologies such as computers and networks [26]. Young adults spend more time in front of screens because they are more likely to use new technologies [22]. In addition to age, gender can also be a determinant; indeed, female workers tend to be more active than their male counterparts because of the double commitment between work and household tasks [27]. Sedentary lifestyles have also been correlated with unhealthy eating habits in teachers, a trend that tends to improve when breaks during working hours increase, emphasizing that breaks during working hours permit this category of workers easier access to healthy foods [16]. Some evidence suggests that physically active teachers are more likely to consume more fruits, vegetables, legumes, grains and olive oil, while less active teachers with a higher BMI consume fewer of these foods [28]. These foods are those typical of the Mediterranean area that are part of a healthy eating pattern known worldwide as the Mediterranean Diet (MD) [29]. Adherence to this type of diet provides significant health benefits, improving emotional and cognitive functioning and physical health [29]. Investigating this aspect is important as teachers are primarily responsible for health promotion among students and serve as role models for the acquisition of healthy lifestyles among students [30]. Lifestyle behaviors such as the practice of physical activity or eating habits and psychological aspects such as physical and mental well-being and their correlations have already been studied in populations similar to ours in other countries [31,32], while in Italy, there was a greater focus on teachers from the university sector before and during lockdown [33,34], or studies focused on individual aspects such as mental [35,36,37] and physical well-being [38]. Thus, little is known about the lifestyle of Italian school teachers. Therefore, the aim of this observational study is to investigate whether there are differences in nursery and primary school teachers’ lifestyles after a COVID-19 lockdown period, and the relationship between perceived physical and mental health and demographics, weight status and lifestyles, such as adherence to the Mediterranean diet (MD) and physical activity level (PAL), in order to identify possible predictors of physical and mental well-being.

## 2. Materials and Methods

### 2.1. Participants and Study Design

A cross-sectional study was conducted with a group of 265 Italian teachers (women 96.2% and men 3.8%) voluntarily recruited from nursery schools (NSs) and primary schools (PSs) during a training course entitled “Natural Moving”, organized by Regional School Office of Sicily (USR), Palermo, Italy, performed in the extracurricular hours from September to November 2021. A total of 382 teachers adhered to data collection, and 265 of them completed all the administered tests. These teachers were divided into NS (43.4%) and PS groups (56.6%). Only teachers with permanent contracts who have been practicing the profession for at least 5 years are included in this study. All participants signed informed consent, voluntarily participating in this study. The investigation was approved by the local ethics committee from the University of Palermo (protocol No. 2/2018) and complies with the criteria on the use of people in research, as stipulated in the Declaration of Helsinki.

An online survey was conducted to collect data, including age, gender, weight, height, school teaching degree (NS and PS), education level (high school and graduation degree). Three questionnaires were administered to analyze the lifestyle determinants, such as physical activity level, adherence to MD and perceived physical and mental health. These questionnaires were compiled as Google forms during video calls using the online platform “Google Meet”, under the supervision of the physical activity (PA) experts. Participants were given as much time as necessary to complete the questionnaires and were asked to keep the video camera on during all the administration.

### 2.2. Tools for Assessing Selected Aspects of Lifestyle and Health 

The PAL of the participants was assessed with the short form of the International Physical Activity Questionnaire (IPAQ-SF) converted to a digital Google form format for online administration [39]. This questionnaire is indicated for adults aged 15–69 years [24,40,41]. Its items explore the duration and frequency of exercise, including walking, moderate and vigorous PA and time spent in sedentary activities, referring to the previous week. The total IPAQ score is calculated as a continuous variable (average minutes/MET per week) or as categories (inactive, sufficiently active and active, highly active levels). An inactive PAL is obtained when an adequate PAL is not attained to fall within the other two levels. A sufficiently active PAL is attained by performing vigorous PA for ≥20 min/day for ≥3 days/week, or through moderate exercise or walking performed for ≥30 min/day, for ≥5 days/week, or again through the combination of moderate or vigorous exercise for ≥5 days/week accumulating at least 600 MET min/week. Active or highly active PA is characterized by ≥3 days per week, with an accumulation of at least 1500 MET min/week, or by cumulative PA for seven days per week of any combination of walking, moderate or vigorous exercise for at least 3000 MET min/week [24]. To calculate the data in our study, we used the automatic IPAQ report created by Di Blasio and colleagues [39].

Dietary patterns were analyzed with the Mediterranean Diet Adherence Screener (MEDAS) questionnaire [42], which assesses the adherence to the MD through 14 items, including the consumption frequency of specific foods typical of the MD, such as fruits, vegetables, olive oil, nuts, fish and wine. All foods that deviate from the MD, such as red meat, carbonated drinks, butter and sweets, were also considered in the questionnaire. Each question was given a score of 0 to 1, so the final score will vary from 0 to 14. According to the literature, we categorized the adherence to MD in this way: weak adherence, ≤5; moderate to fair adherence, 6–9; good or very good adherence, ≥10 [43].

To investigate levels of perceived health, we administered the SF-12 questionnaire (short form of the original SF-36 Health Survey) [44] converted to a digital Google form for online administration. This is a psychometric tool that with two indices, concerning the Physical Component Summary (PCS) and Mental Component Summary (MCS) assessing 8 items on physical functioning, functional limitations due to physical and emotional health problems, freedom from bodily pain, perception of general health, vitality, social functioning and mental health, the physical and mental state of participants. In other terms, it is a tool that allows us to assess the health status and health-related quality of life of particular populations. For the calculation of indices, we used a Microsoft Excel spreadsheet made by Ottoboni et al. [45], while for the evaluation of these parameters, we referred to the normative values divided by age and gender given by Schupp et al. [46].

### 2.3. Statistical Analysis

To identify an appropriate sample size for recruitment, the G-power software (G*Power version 3.1.9.4, using Power 1-β err prob ≥ 0.80, Effect Size = 0.3, α err prob = 0.05) was used, which defines the minimum number of participants needed for the intended research model.

An exploratory analysis was initially carried out, and participant’s characteristics were expressed as means, standard deviations and percentage values. The quantitative variables present in the analysis were age, weight, height and body mass index (BMI), and the two components of the SF-12 questionnaire (PCS and MCS) were total MET min/week and MD adherence score. The qualitative variables were educational levels (secondary school and graduation degree), weight status (normal, overweight and obese), PAL (inactive, sufficiently active and active or highly active). These variables were dichotomized before the analysis.

The normal distribution of the data was verified with the Kolmogorov–Smirnov test. In the inferential statistical analysis, when variables were normally distributed, we used a *t*-test or One-Way ANOVA test to assess the differences between the means of the examined variables. When they were not normally distributed or qualitative, we used a non-parametric Mann–Whitney U test or Kruskal–Wallis test to assess the differences. For a more detailed analysis, we performed a stratification by PALs (inactive, sufficiently active and active/very active), weight status (normal, over and obese) and school teaching grade (NS and PS). A correlation analysis between the qualitative binary variables was performed using a non-parametric correlation test (Sperman’s rho). Afterwards, a multinomial logistic regression analysis reporting Odds Ratios (ORs) and 95% Confidence Intervals (CIs) was conducted among the qualitative variables in order to evaluate associations between the PCS and MCS (which were considered the dependent variables) and the independent variables (school teaching grade, education, age, weight status, MD adherence, PAL).

The *p*-value significance level was set at 0.05.

All analyses were performed with Jamovi software (The Jamovi project (2021) Version 1.8.0.1 (Computer Software), Sydney, Australia, retrieved from https://www.jamovi.org accessed on 8 December 2023).

## 3. Results

A descriptive analysis of the quantitative variables of the sample in their totality (TS) and in the two groups of NS and PS teachers is shown in Table 1. The total MET score derived from the IPAQ questionnaire was high in both groups, with PS teachers having a significantly higher total MET compared to the NS group (*p* = 0.046). This difference was also present when the corresponding categorical classes (PAL) were considered (Table 2). Furthermore, the MD adherence was from moderate to fair in both groups, and it was not significantly different between them. The results of the SF-12 questionnaire showed that PS teachers had a higher PCS score (*p* = 0.030) compared to NS teachers, and this difference persists when dichotomized variables are considered (Table 2); on the contrary, both groups exhibited an MCS score above average, comparing these data with normative references to different age classes [46], but showed no between-groups differences.

Table 2 shows descriptive statistics for qualitative binary variables. The results indicated a significantly higher percentage of teachers with an educational secondary level in the PS (*p* < 0.001) compared to the NS and a significantly higher percentage of teachers with a high graduation degree level in the NS than the PS (*p* < 0.001). Stratifying the sample according to educational levels (see Appendix A) in relation to PALs, no statistical differences were found.

Regarding the weight status, both groups showed a higher percentage of normal-weight teachers compared to their overweight and obese counterparts, and no difference between groups was found.

The total sample of teachers with an active or highly active level had a significantly bigger score that ranged from fair to moderate in MD adherence and PCS variables in comparison with sufficiently active (*p* = 0.023) and inactive (*p* < 0.001) teachers (see Table 3). Conversely, no significant difference was found in the MSC score according to the level of physical activity. In detail, the NS group with an active/highly active level exhibited a significant increase in MD adherence compared with sufficiently active (*p* = 0.030) and inactive (*p* < 0.001) teachers, while no significant difference was observed among the PS teachers with different levels of physical activity, despite their level of MD adherence being from fair to moderate. Comparing NS and PS teachers, we noted that NS teachers with an active or highly active level had an MD adherence score significantly higher than PS teachers (*p* = 0.015).

NS teachers with an active or highly active level displayed a significantly higher PCS score than their inactive counterparts (*p* = 0.016) and a sufficient level of physical health, whereas the MCS score was the same among the NS teachers, independently by the level of physical activity. The PCS score of the PS teachers did not significantly change according to the physical activity level.

Stratifying the sample according to weight status (see Appendix A), the only statistically significant difference was found to be for the PCS score in the NS teachers between normal weight and obese groups (*p* = 0.041), whereas, when stratifying the sample according to educational levels (see Appendix A), only the MCS score in the NS showed a significant difference between elementary and middle/high school levels, with *p* = 0.008. Appendix A due to the few significant observed. Stratification by gender (F/M) was not taken into consideration since men were only 3.8%.

The correlation analysis between binary variables is shown in Table 4. Even though low values of Spearman correlation coefficients were found, PCS was significantly correlated to the following: school teaching grade, with primary teachers showing high perceived physical health (*p* < 0.05); weight status, with overweight or obese teachers perceiving lower PCS (*p* < 0.05); MD adherence, with those adherents to MD having higher physical well-being (*p* < 0.05); PAL, with active teachers showing higher PCS (*p* < 0.01). For mental health, only weight status was found to be correlated to MCS, with overweight/obese teachers showing lower mental well-being (*p* < 0.05).

The logistic regression analysis results for PCS and MCS are shown in Table 5.

Active teachers are more likely to perceive good physical health (OR 2.10, 95%CI 1.05–4.2, *p* = 0.035). Even though not statistically significant, a trend exists (*p* < 0.10) for school teaching grade and weight status: primary schoolteachers have a higher probability of physical well-being (OR 0.48, 95%CI 0.33–0.78 *p* = 0.020) compared to nursery school teachers; overweight and obese teachers have almost double the risk of perceiving low physical health (OR 0.59, 95%CI 0.33–1.03, *p* = 0.062) compared to normal-weight teachers.

With regard to mental health, the only predictor of MCS was confirmed to be weight status, with overweight and obese teachers showing a double probability of perceiving low mental health (OR 2.18, 95%CI 1.13–4.18, *p* = 0.020). A trend is shown for active teachers who have almost two-times the odds of perceiving good mental health (OR 1.89, 95%CI 0.93–3.83, *p* = 0.077).

## 4. Discussion

In this study, we aimed to investigate the differences in the lifestyle habits of nursery and primary school teachers, focusing on their perceived physical and mental health in relation to sociodemographic aspects, weight status, PALs and MD adherence in the months following the COVID-19 lockdown. Knowing these aspects is important to be able to identify whether or not the risk factors for the onset of chronic non-communicable diseases associated with lifestyle determinants have changed as a result of the lockdown. The aim of acquiring these data is to improve the quality of life of this category of workers by identifying appropriate intervention strategies. Therefore, we carried out this cross-sectional study to investigate these aspects that contribute to determining the lifestyle of this class of workers, as previously shown in the literature [28]. According to Gomes et al. [47], teachers’ positive or negative experiences with physical activity can influence their propensity to create an educational climate that promotes PA and active lifestyles in schools.

In our study, 83.4% of teachers showed a level of physical activity ranging from sufficiently active to active or very active. This high percentage is in contrast to those reported in several studies conducted in other countries on samples of public-school teachers before [22,24,48] and after lockdown [32]. In fact, these studies emphasize the wide prevalence of sedentary behaviors in this category of workers and show that inadequate levels of physical activity [24] predispose one to an increase in cardiovascular risk factors [22] and the appearance of musculoskeletal disorders with the onset of chronic pain [48]. These findings suggest the need to promote PA both during and outside working hours. In particular, we found a significantly higher PAL in primary school than nursery teachers. These results could be explained by the kind of work activity performed. According to some evidence in the literature, this category of worker may be among those engaged in non-sedentary work activities, considering that 95% of work activities are conducted in the orthostatic position, thus maintaining a prolonged isometric contraction to oppose the force of gravity [22,49]. This may explain the high levels of physical activity of the teachers in our study. Moreover, our results might be explained by the high percentage of women (96.2%) present in our sample, who are considered to be more active compared with men as they are also employed in carrying out domestic activities [27]. Our sample can be considered representative of the reference professional category; indeed, in Sicily, among 10,861 nursery school teachers, only 142 are men, and among 25,949 primary school teachers, only 1154 are men. This datum indicates that there is a social propensity of women and men to prefer or not prefer this professional career [50].

The prevalence of women in this occupational field is a peculiar feature and is attributed to the tendency, since the 20th century, for women to enter the workforce, during a historical period when teaching was regarded as an extension of domestic activities [27]. Furthermore, higher levels of physical activity, especially in women, were associated with a higher work capacity [51], so in this regard, our results are encouraging. Nevertheless, it should also be considered that, in our study, the teachers, by voluntary joining a training course on the promotion of physical activity, actually expressed a personal interest in this educational content that might have influenced our results. However, all teachers included were from educational backgrounds, different in degrees of exercise and sports sciences.

A recent study reported a negative relationship between BMI and food intake in school teachers. In particular, less physically active teachers had unhealthy food habits [28]. The COVID-19 pandemic seems to have influenced this population of workers by favoring or hindering the intake of unhealthy foods [2]. In our study, in both groups (NS and PS), almost 30% of teachers were overweight and obese, and BMI was negatively correlated with moderate physical activity (MA). These results are confirmed by other studies [22,25,51].

Regarding the relationship between PALs and MD adherence, in our study, teachers with a level of active or highly active PA had moderate to fair adherence to the Mediterranean Diet (score ≥ 9) compared to those who were sufficiently active or inactive in all groups with a lower MD adherence group (score < 9) (Table 3). This is in agreement with the results of other studies, in which teachers who adopted sedentary behaviors had unhealthy eating patterns compared to their more active colleagues [2,16,50]. It should be emphasized that the results of adherence to the Mediterranean Diet achieved by our sample were sufficient, and these results could explain the high percentage of overweight and obese individuals. Indeed, the normal-weight teachers tended to have better adherence to the Mediterranean Diet compared with their overweight and obese counterparts. These findings are confirmed by recent studies on this population that suggest increasing levels of physical activity, being careful with eating habits and proposing strategies for reducing levels of work-related stress [8,52].

A recent study evaluated the effect of teachers’ prejudice on diet and exercise and how teachers’ opinions can influence children’s behaviors. Nevertheless, these results might be hindered by the health level of teachers, which greatly affects the quality of teaching and, thus, the academic performance of students and their learning [53].

Comparing the two groups of nursery and primary teachers, statistically significant differences were found for the variables Total MET, PCS and educational levels. In particular, PS teachers presented higher levels in energy expenditure, perceptions of their own physical health and education compared to NS. Due to the lack of papers in the literature on the comparison of these two groups, it is difficult to provide argued comments on this issue. We have found just one study targeted on nursery school teachers assessing PA level by gender [54], where higher levels of PA in male workers have been demonstrated, suggesting that these could positively affect the PA of children. Thus, to our best knowledge, our study might be the first to assess the possible differences between these two groups of workers and, therefore, provide a new contribution to the scientific literature.

Moreover, primary school teachers showed a correlation with perceived physical health, and although this relationship was not significant in the logistic regression, a trend remained. We hypothesize that the difference concerning Total MET and PCS between the groups might be due to the higher school activity level of PS teachers than NS counterparts during the lockdown period, as online lessons were envisaged for the primary school sector and not for the nursery.

In our study, the primary school teachers had higher levels of education than the nursery school group; however, no statistically significant difference emerged when stratifying the sample by levels of education (secondary and degree) in relation to the PALs. Pirzadeh et al. also found no correlations between education levels and PA levels, suggesting that this result could be explained by a lack of specific knowledge on the topic [55]. Not having specific knowledge about a healthy lifestyle might actually limit its dissemination by teachers and, at the same time, influence one’s lifestyle significantly [14]. Considering that our population of teachers was formed in other educational fields and not exercise and sport sciences, the lack of specific knowledge might have influenced the results of the IPAQ and SF-12 questionnaires, leading to an overestimation of physical activity levels as well as physical and mental health. In addition, regarding the education level, a recent study [56] on preschool teachers found that the level of education possessed by teachers could influence the PA level of teachers, emphasizing that teachers could be role models for children.

The analysis of possible factors correlated to perceived physical and mental health revealed that weight status was significantly correlated to both PCS and MCS, with overweight/obese teachers perceiving lower physical and mental health; when these variables were included in a logistic regression model, the association was not confirmed for PCS, even though a kind of trend exists, while it was confirmed for MCS, suggesting that maintaining or reaching an optimal weight status could significantly influence the psychological health status and, in particular, mental well-being.

PCS was significantly correlated to MD adherence in the correlation analysis, with those who were not adherent to MD having lower physical well-being, but this association was not confirmed in the regression model.

PCS was finally correlated to PAL, and this relation was highly significant in the regression model, with active teachers having a double probability of perceiving their higher physical health. Even though the association between MCS and PAL was not evidenced, a trend exists. This result could be explained, in contrast with the previous literature, by considering teachers’ mental well-being as the result not only of physical activity levels and eating habits but also as the result of factors such as the quality of the work environment [57]. PA appears to be able to positively influence teachers’ perceptions on physical and mental well-being, and, conversely, teachers with healthy eating habits tended to carry out a greater amount of weekly PA and perceive greater physical health [58].

The results obtained from the logistic regression analysis confirm that more active teachers tend to perceive greater physical and mental well-being, regardless of their BMI. Our results are, in part, in line with other studies, where factors, such as teachers’ age, PALs and diet, are associated with perceived levels of physical and mental well-being [59]. In several studies, the physical and mental well-being of teachers is rated lower than that of the general population due to the levels of physical and mental stress associated with the work activity conducted [59,60]. Higher PALs are associated with better perceptions of their own well-being, but in the case of female teachers, the contrary is also true. Indeed, the presence of dual commitments, work and domestic duties negatively affects the perception of overall health, predisposing to a higher level of stress and increased absenteeism compared to male colleagues [58]. In addition, increased levels of stress, anxiety, depression and sleep disorders among teachers also appear to have occurred as a result of the COVID-19 pandemic [3,9].

These results together confirm the need for increasing physical activity interventions targeted to teachers and promoting physical activity practice as a well-being means to achieve optimal psychological health status.

A limitation of this study is that the use of a self-reported questionnaire to collect information on the amount of physical activity performed by teachers has objective limitations that could be overcome by using inertial accelerometers. Also, weight and height that were self-reported and not directly measured contribute to limiting the study results, since the BMI and weight status categories could be biased. Moreover, another limit is that the sample was almost entirely composed of women. In addition, the sample size should be increased by extending the study to the whole national territory.

## 5. Conclusions

In consideration of the importance of the institutional role occupied by preschool and primary school teachers, assessing all those aspects that may affect the welfare of this category of workers is notable. Psychological factors may influence the implementation of sedentary behaviors, and the food choices of teachers should be considered in the intervention planning to motivate and promote the acquisition or maintenance of healthy lifestyles. Weight status seems to improve physical and mental well-being. Increasing knowledge and awareness about exercise and nutrition in this school population might contribute to a change from unhealthy lifestyles. A conspicuous body of literature has focused on health promotion interventions for students, but these interventions do not specifically include teachers. The results of the present study give hope because they show a category of workers with acceptable levels of physical and mental health, despite the difficult historical period experienced. It would be advisable to provide workplace health promotion interventions that would include both students and teachers. Integrating teachers into health promotion interventions in schools could lay the foundation for creating an educational climate that promotes healthier lifestyles.

## Figures and Tables

**Table 1 jfmk-09-00033-t001:** Descriptive statistics of the quantitative considered variables in the total sample (*n* = 265) and in the nursery (*n* = 115, 43.4%) and primary (*n* = 150, 56.6%) school groups.

	TS ^a^	NS ^b^	PS ^c^
Age (years)	49.22 ± 6.95	49.0 ± 6.65	49.4 ± 7.19
Weight (Kg)	64.11 ± 10.28	63.5 ± 9.60	64.6 ± 10.8
Height (cm)	164.22 ± 6.7	163 ± 6.06 *	165 ± 7.07 *
BMI ^d^ (Kg/m^2^)	23.77 ± 3.54	23.8 ± 3.47	23.7 ± 3.60
Total MET ^e^ (score)	2888.0 ± 2604.0	2678.0 ± 2736.0	3049.0 ± 2495.0 *
MD ^f^ adherence (score)	8.86 ± 1.90	9.05 ± 1.95	8.71 ± 1.87
PCS ^g^ (score)	49.65 ± 7.50	48.51 ± 7.77	50.53 ± 7.18 *
MCS ^h^ (score)	51.64 ± 8.05	52.11 ± 7.61	51.28 ± 8.38

Data are presented as means ± SD; ^a^, TS, Total Sample; ^b^, NS, Nursery School; ^c^, PS, Primary School; ^d^, BMI, body mass index; ^e^, Metabolic Equivalent of Task; ^f^, Mediterranean Diet; ^g^, Physical Component Summary; ^h^, Mental Component Summary; * *p* < 0.05; NS vs. PS.

**Table 2 jfmk-09-00033-t002:** Descriptive statistics of the categorized binary variables in the total sample (N = 265) and in the nursery (*n* = 115, 43.4%) and primary (*n* = 150, 56.6%) school groups.

		TS ^a^		NS ^b^		PS ^c^	
		N	%	N	%	N	%
Age classes	Younger (<50 years)	140	52.8	65	46.4	75	53.6
	Older (≥50 years)	125	47.2	50	40	75	60
Weight status	Normal	190	71.7	79	41.6	111	58.4
	Overweight/obese	75	28.3	36	48	39	52
Educational level	Secondary	132	49.8	61	46.2	71	53.8 ***
	Graduate	133	50.2	71	53.4 ***	62	46.6
PAL ^d^	Inactive	44	16.6	23	52.3	21	47.7 *
	Active/very active	221	83.4	92	41.6	129	58.4
MD ^e^ adherence	Non adherent	170	64.2	69	40.6	101	59.4
	Adherent	95	35.8	46	48.4	49	51.6
PCS ^f^	Low	107	40.4	54	50.5	53	49.5 *
	High	158	59.6	61	38.6	97	61.4
MCS ^g^	Low	80	30.2	34	42.5	46	57.5
	High	185	69.8	81	43.8	104	56.2

Data are presented as numeric and percentage frequencies; ^a^, TS, Total Sample; ^b^, NS, Nursery School; ^c^, PS, Primary School; ^d^, PAL, Physical Activity Level; ^e^, MD, Mediterranean Diet; ^f^, PCS, Physical Component Summary; ^g^, MCS, Mental Component Summary; * *p* < 0.05, *** *p* < 0.001, NS vs. PS.

**Table 3 jfmk-09-00033-t003:** Mediterranean Diet adherence, perceived physical health and mental health by physical activity level and education teaching grade classes.

	Inactive			Sufficiently Active			Active or Highly Active		
Scores	TS ^a^	NS ^b^	PS ^c^	TS ^a^	NS ^b^	PS ^c^	TS ^a^	NS ^b^	PS ^c^
MD ^d^ adherence	8.1 ± 1.7 **	8.0 ± 1.5 **	8.1 ± 1.9	8.6 ± 2.0 °	8.8 ± 2.1 °	8.4 ± 1.9	9.3 ± 1.8 **^,^°	9.9 ± 1.7 **^,^°^,#^	9.0 ± 1.7 ^#^
PCS ^e^	46.3 ± 8.5 **	44.3 ± 8.4 *	48.5 ± 8.1	49.12 ± 7.8 °	49.1 ± 7.5	51.2 ± 8.0	51.3 ± 6.4 **^,^°	50.24 ± 7.02 *	51.9 ± 5.9
MCS ^f^	49.9 ± 9.3	50.6 ± 8.6	49.0 ± 10.2	51.4 ± 8.5	49.2 ± 8.2	51.5 ± 9.0	52.4 ± 7.1	53.6 ± 6.5	51.7 ± 7.4

Data are presented as means ± SD; ^a^, TS, Total Sample; ^b^, NS, Nursery School; ^c^, PS, Primary School; ^d^, Mediterranean Diet; ^e^, PCS, Physical Component Summary; ^f^, MCS, Mental Component Summary; * *p* < 0.05; ** *p* < 0.01, in Total sample, NS and PS, Inactive vs. Active/Highly Active, while ° *p* < 0.05; Sufficiently Active vs. Active/Highly Active; ^#^
*p* < 0.05; NS vs. PS for Active/Highly Active.

**Table 4 jfmk-09-00033-t004:** Spearman’s correlation coefficients between perceived physical/mental health and the considered study variables in the sample of schoolteachers (N = 265).

	PCS ^a^ (Low–High)	MCS ^b^ (Low–High)
	Spearman’s Rho	
School teaching grade (nursery-primary)	0.117 *	−0.012
Educational Level (secondary-graduate)	−0.038	0.031
Age (younger-older)	0.069	−0.004
Weight status (normal-overweight/obese)	−0.132 *	−0.139 *
MD ^c^ adherence (non adherent-adherent)	0.118 *	0.029
PAL ^d^ (inactive-active/very active)	0.170 **	0.104

^a^, PCS, Physical Component Summary; ^b^, MCS, Mental Component Summary; ^c^, MD, Mediterranean Diet; ^d^, PAL, Physical Activity Level; * *p* < 0.05; ** *p* < 0.01.

**Table 5 jfmk-09-00033-t005:** Logistic regression results on PCS and MCS in relation to all qualitative variables.

	PCS ^a^ (Base Outcome Low vs. High)			MCS ^b^ (Base Outcome Low vs. High)		
	OR ^c^	95% CI ^d^	*p*-Value	OR ^c^	95% CI ^d^	*p*-Value
School teaching grade						
base outcome nursery vs. primary	1.59	0.93–2.72	0.092	0.98	0.55–1.74	0.948
Education						
base outcome primary/secondary vs. graduate	1.06	0.60–1.85	0.848	1.09	0.61–1.95	0.763
Age						
base outcome younger vs. older	1.03	0.99–1.07	0.124	0.92	0.54–1.60	0.783
Weight status						
base outcome normal vs. overweight/obese	0.59	0.33–1.03	0.062	0.48 *	0.33–0.78	0.020
MD ^e^ adherence						
base outcome non adherent vs. adherent	1.47	0.85–2.56	0.170	1.10	0.62–1.96	0.735
PAL ^f^						
base outcome inactive vs. active/very active	2.10 *	1.05–4.20	0.035	1.89	0.93–3.83	0.077

^a^, PCS, Physical Component Summary; ^b^, MCS, Mental Component Summary; ^c^, OR, Odds Ration; ^d^, CI, Confidence Interval; ^e^, MD, Mediterranean Diet; ^f^, PAL, Physical Activity Level; * *p* < 0.05.

## Data Availability

The data presented in this study are available on request from the corresponding author.

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
