# Peer review of "Relationship between Lifestyle Determinants and Perceived Mental and Physical Health in Italian Nursery and Primary School Teachers after the COVID-19 Lockdown"

_jfmk, 2024, doi:10.3390/jfmk9010033_

Round 1

Reviewer 1 Report

Comments and Suggestions for Authors

11.    In the legends of all tables, it should be mentioned data or variables are expressed as means ± SE or SD (standard error or Standard deviation).

2 2. In discussion, page 8, line 310 - 312.

  Pasting from the text:

 “Regarding the relationship between PALs and MD-Adherence, in our study the teachers with a level of active or highly active PA had a moderate to fair adherence to the Mediterranean diet compared to those who were sufficiently active or inactive in all groups. 

 This paragraph is not completely clear, First, a phrase should be added so that the whole paragraph so that it becomes:  

Regarding the relationship between PALs and MD-Adherence, in our study the teachers with a level of active or highly active PA had a moderate to fair adherence to the Mediterranean diet (≥ 9) compared to those who were sufficiently active or inactive in all groups with lower MD-Adherence group (less than 9) table 3.

 3. Abstract: This above-mentioned results about higher observed MD-Adherence score in physically active or highly active participants in groups should be incorporated/expressed in the abstract.

33. Table 4 : Cosmetically-speaking and for clearer presentation the table should be laid out/presented in in page 7 without separation of  two lines of it in page 6 and the rest in page 7, specially for readers or students/researchers who use printed hard copies. Even for readers of the screen could be distracted when they want to focus on this table of correlations. Therefore, considering table 4 in one same page should be seriously considered.

44.     It should be mentioned either in abstract (line 15) or introduction (line 94) or both that the aim of this observational study was to…..etc. i.e. observational study should be clearly stated.

5 5.       Reducing the introduction by few lines to be tailored to the directly related points of the study may be a constructive positive suggestion for a stream-lined draft/story.

In this context, no need to talk about cardiac cancer disease unless very briefly and not repeatedly as no questionnaire or related variable was included in the study.

Comments on the Quality of English Language

The manuscript is written in excellent English. However recommended scientific reviews or minor redraft or additions, as mentioned above, are needed to make those sentences clear to the point. But in general the whole manuscript is written in very well clear and English well-structured text. 

Reviewer 2 Report

Comments and Suggestions for Authors

Dear Authors,  

1) Providing body mass and height in the ‘Abstract’ section is unnecessary. An average BMI value may be considered.

2) The ‘Abstract’ does not provide any results taking the analysed variables into account using the indicated statistical tests. In my opinion, this section needs to be completed.

3) In the ‘Introduction’, the objective should be extended and clarified, e.g. in the form of research questions.

4) I wonder about the purpose of Table 1, since the most important data regarding BMI are also in Table 2. Would it be possible to include the average BMI value in Table 2? Data on BM and H are less important (in this work, they are only important for calculating BMI). Please, consider.

5) Lack of formulated criteria for inclusion and exclusion from the group.

6) Sex (approx. 4% M), which makes it impossible to distinguish the gender variable, seems to be one of the ‘Limitations’. Similarly, online data collection, declared results, e.g. body mass, are also examples of ‘Limitations’.

7) Section 2.2. should be titled in such a way as to indicate that it concerns the applied research tools, e.g. ”Tools for assessing selected aspects of lifestyle and health”.

8) Please complete the reliability indicators of the tools used to assess the implementation of the Mediterranean diet and SF-12.

9)What key was used in the questionnaire to assess MD (since recommended and non-recommended products were included, was reverse scoring used for non-recommended products in the MD diet)? Please, explain.

10) In my opinion, the Tables (titles and legends) should be more precise. Some tables do not provide information on what descriptive statistics are presented, on the basis of what statistical tests specific results were obtained, etc.

11) Please check the consistency of the data in the Tables with the description (especially Table 2).

12) If we provide exact p values in the text, then we use the = sign, and the < sign if we assume, for example, less than 0.05 or p<0.001. This remark applies to the entire work.

13) I suggest supplementing the conclusions with more specific findings of cognitive nature, directly related to the purpose of the research. The current version of the applications has a more applied dimension (also very important).

Comments on the Quality of English Language

The English language in the text is for the most part, correct. However, minor revisions are required due to, e.g. lacking prepositions, mistakes in article usage, repetitions, usage of passive voice, etc. I would recommend the manuscript be checked by a native speaker.

Round 2

Reviewer 2 Report

Comments and Suggestions for Authors

Dear Authors,

Thank you for the corrections that allowed to improve the quality of the manuscript. Unfortunately, there are still errors that need to be addressed.

1) I do not understand how to count the percentage of NS and PS teachers in Table 2. Each of these distinguished groups (NS and PS) should be counted separately, in relation to their number, i.e. 115 and 150 people).

2) It is necessary to thoroughly verify the consistency of the results included in the tables with their description, as I still notice inaccuracies.

For example, in the sentence "For mental health, only weight status was found to be correlated to MCS, with overweight/obese teachers showing a higher mental well-being (p<0.05)", according to Table 4, it should be "lower" and not “higher” (Rho is -0.139*). The direction (sign) of the correlation in MCS and PCS is the same.

Another example concerns Table 5. The sentence “With regard to mental health, the only predictor of MCS was confirmed to be weight status, with overweight and obese teachers showing a double probability of perceiving a high mental health (OR 2.18, 95%CI 1.13- 4.18, p=0.020)”; according to Table 5, this is not correct.

According to the data from Table 5, it should be "...a low mental health (OR 0.48; 95% CI 0.33-0.88; p=0.020)”. Please, verify and correct.

If there are errors in the ‘Results’ section, they may also appear in the ‘Abstract’, ‘Discussion’ and ‘Conclusions’ sections, thus, it is necessary to check the entire work very carefully for the correctness and consistency of the data.

3) The ‘Conclusions’ still need to be reformulated. In the initial part of the conclusions, there is reference to the literature, i.e. suitable for an introduction or discussion section. There are no cognitive conclusions directly resulting from the research. There are only assumptions of applicative nature.

4) I would prefer a proofreading version with changes marked by colour (without deletions in the track changes option to make the text more readable).

Comments on the Quality of English Language

Dear Editor,

The authors have made significant corrections, but unfortunately, there are still errors and inaccuracies that require very thorough verification and correction, 

kind regards, 

Maria Gacek

Round 3

Reviewer 2 Report

Comments and Suggestions for Authors

In the description of Tab. 5, the text has been corrected, but the OR in brackets is still incorrect (OR 2.18; 95%CI 1.13-4.18; p=0,020), and OR is given in the table (OR 0.48; 95%CI 0.33-0.78; p=0.020). The OR given in brackets in the text should be consistent with that provided in the table. See, it's correct in the abstract.
